# Reduction of Tumor Growth with RNA-Targeting Treatment of the NAB2–STAT6 Fusion Transcript in Solitary Fibrous Tumor Models

**DOI:** 10.3390/cancers15123127

**Published:** 2023-06-09

**Authors:** Yi Li, John T. Nguyen, Manasvini Ammanamanchi, Zikun Zhou, Elijah F. Harbut, Jose L. Mondaza-Hernandez, Clark A. Meyer, David S. Moura, Javier Martin-Broto, Heather N. Hayenga, Leonidas Bleris

**Affiliations:** 1Department of Bioengineering, University of Texas at Dallas, Richardson, TX 75080, USA; yxl121030@utdallas.edu (Y.L.); john.nguyen12@utdallas.edu (J.T.N.); manasvini30@gmail.com (M.A.); zikun.zhou@utdallas.edu (Z.Z.); clark.meyer@utdallas.edu (C.A.M.); 2Center for Systems Biology, University of Texas at Dallas, Richardson, TX 75080, USA; 3Department of Chemistry and Biochemistry, University of Texas at Dallas, Richardson, TX 75080, USA; elijah.harbut@utdallas.edu; 4Health Research Institute Fundacion Jimenez Diaz, Universidad Autonoma de Madrid (IIS/FJD-UAM), 28049 Madrid, Spain; jmondaza@atbsarc.org (J.L.M.-H.); dmoura@atbsarc.org (D.S.M.); jmartin@atbsarc.org (J.M.-B.); 5University Hospital General de Villalba, 28400 Madrid, Spain; 6Medical Oncology Department, University Hospital Fundación Jimenez Diaz, 28040 Madrid, Spain; 7Department of Biological Sciences, University of Texas at Dallas, Richardson, TX 75080, USA

**Keywords:** solitary fibrous tumor (SFT), CRISPR, antisense oligonucleotides (ASOs), RfxCas13d (CasRx)

## Abstract

**Simple Summary:**

Solitary fibrous tumor (SFT) is a soft-tissue sarcoma occurring in adults and infants. This nonhereditary cancer is the result of an environmental intrachromosomal gene fusion between NAB2 and STAT6 on chromosome 12. Either surgery or radiation is the first line of treatment for this cancer; however for many, this becomes challenging, as the cancer can travel to inoperable areas or reoccur in locations already irradiated. Currently there is no approved chemotherapy regimen for SFTs. Anti-angiogenic drugs developed to treat other cancers have been used on SFTs with limited success. Therefore, there is a need for systemic therapy. In this study, we showed that RNA-targeting technologies (antisense oligonucleotides and CRISPR/CasRx) can be used to specifically suppress the expressions of NAB2–STAT6 fusion transcripts, but not wild type STAT6, and reduce cell proliferation and tumor growth. These results provide the foundation for a potentially novel therapeutical strategy for SFTs.

**Abstract:**

Solitary fibrous tumor (SFT) is a rare soft-tissue sarcoma. This nonhereditary cancer is the result of an environmental intrachromosomal gene fusion between NAB2 and STAT6 on chromosome 12, which fuses the activation domain of STAT6 with the repression domain of NAB2. Currently there is not an approved chemotherapy regimen for SFTs. The best response on available pharmaceuticals is a partial response or stable disease for several months. The purpose of this study is to investigate the potential of RNA-based therapies for the treatment of SFTs. Specifically, in vitro SFT cell models were engineered to harbor the characteristic NAB2–STAT6 fusion using the CRISPR/SpCas9 system. Cell migration as well as multiple cancer-related signaling pathways were increased in the engineered cells as compared to the fusion-absent parent cells. The SFT cell models were then used for evaluating the targeting efficacies of NAB2–STAT6 fusion-specific antisense oligonucleotides (ASOs) and CRISPR/CasRx systems. Our results showed that fusion specific ASO treatments caused a 58% reduction in expression of fusion transcripts and a 22% reduction in cell proliferation after 72 h in vitro. Similarly, the AAV2-mediated CRISPR/CasRx system led to a 59% reduction in fusion transcript expressions in vitro, and a 55% reduction in xenograft growth after 29 days ex vivo.

## 1. Introduction

Solitary fibrous tumor (SFT) is a rare soft-tissue tumor of mesenchymal origin that accounts for less than 2% of all soft tissue sarcomas. In the latest WHO classification published in April 2020, SFTs are subdivided into three categories: benign (locally aggressive), NOS (rarely metastasizing), and malignant [1,2,3]. Traditionally, these tumors present a unique diagnostic challenge due to their gross and histologic features overlapping with many other soft-tissue tumors. A breakthrough occurred in 2013, when all SFTs were found to have a version of hallmark intrachromosomal gene fusion between NAB2 and STAT6 on chromosome 12 [4,5]. Research suggests that NAB2–STAT6 is the oncogenic driver [4,6,7]. Normally, early growth response-1 (EGR-1) activates NAB2, and NAB2 in turn represses oncogenic EGR-1 target genes [4]. However, in the case of the NAB2–STAT6 fusion, the activation domain of STAT6 replaces the repression domain of NAB2. As a result, instead of repressing EGR-1 target genes, the fusion protein activates them [8]. Despite these advances in the classification and understanding of the molecular mechanisms of SFTs, no SFT-specific systemic therapeutic options are yet available [9]. Anti-angiogenic drugs developed to treat other cancers have been used on SFTs with limited success [10]. None of the chemotherapies enables complete remission, with the best response being a partial response or stable disease for several months.

ASOs (antisense oligonucleotides) are synthetic single-stranded oligonucleotides or oligonucleotide analogs (typically 15–25 bp in length) which can bind to RNAs via Watson–Crick base pairing. Both protein-coding RNAs (messenger RNAs, mRNAs), noncoding RNAs such as long noncoding RNAs (lncRNAs), and microRNAs can be targeted via distinct mechanisms, including the RNase H-dependent and RNase H-independent pathways [11,12,13]. Various chemical modification methods, such as phosphorothioate and 2′-*O*-methoxyethyl (MOE) modification, are commonly used to enhance the binding affinity of ASOs to target RNAs, increase metabolic stability, and decrease possible adverse effects. To date, eight ASOs have been approved by the FDA for disorders including Duchenne muscular dystrophy, spinal muscular atrophy, and cytomegalovirus (CMV) retinitis [14,15,16,17,18,19,20,21,22].

CRISPR (clustered regularly interspaced palindromic repeats) technology has revolutionized the field of genetic engineering [23,24,25,26]. Various CRISPR-mediated DNA editing technologies, such as the type II CRISPR/SpCas9 system or type V CRISPR-Cas12 (Cpf1) system, have been extensively studied and already moved into clinical trials [27,28,29]. In parallel, efforts have been made to discover and establish diverse Cas effectors which can precisely manipulate RNA molecules. More recently, the Type VI CRISPR/RfxCas13d (CasRx) system, which is derived from *Ruminococcus flavefaciens* XPD3002 and possesses dual RNase activities, was found to efficiently process target RNAs in mammalian cells with fewer non-specific targeting effects compared to other RNA-editing technologies such as shRNAs (short hairpin RNA) [30,31,32].

We note that all NAB2–STAT6 fusion types in SFTs create unique fusion RNA transcripts which are distinct from wild type NAB2 or STAT6 transcripts. Therefore, this study investigated both ASO- and CasRx-based RNA targeting technologies to specifically suppress the expression level of NAB2–STAT6 fusion transcripts, which we hypothesized would exert anti-tumor benefits for SFTs. We report suppression of RNA fusion transcripts and associated reduction in growth rates in vitro and in xenograft models, pointing to a potentially viable therapeutical strategy for SFTs.

## 2. Materials and Methods

### 2.1. Mammalian Cell Culture

All cells investigated herein, that is HCT116 (American Type Culture Collection, Manassas, VA, USA catalog number: CCL-247), NS-poly, NS-11, NS-17, and NS-23, were maintained at 37 °C, 100% humidity and 5% CO_2_. The cells were grown in Dulbecco’s modified Eagle’s medium (DMEM media, Invitrogen, Waltham, MA, USA, catalog number: 11965-1181) supplemented with 10% fetal bovine serum (FBS, Invitrogen, catalog number: 26140), 0.1 mM MEM non-essential amino acids (Invitrogen, catalog number: 11140-050), and 0.045 units/mL of Penicillin and 0.045 units/mL of Streptomycin (Penicillin-Streptomycin liquid, Invitrogen, catalog number: 15140). To pass the cells, the adherent culture was first washed with PBS (Dulbecco’s Phosphate Buffered Saline, Mediatech, Manassas, VA, USA, catalog number: 21-030-CM), then trypsinized with Trypsin-EDTA (0.25% Trypsin with EDTAX4Na, Invitrogen, catalog number: 25200) and finally diluted in fresh medium. Additionally, for maintaining NS-poly cells, hygromycin (200 µg/mL, Thermo Fisher Scientific, Waltham, MA, USA, catalog number: 10687010) was included in the complete growth medium.

### 2.2. Long-Range Genomic PCR and RT (Reverse Transcription)-PCR

For long-range genomic PCR, total genomic DNAs were harvested using DNeasy Blood & Tissue kit (Qiagen, Germantown, MD, USA, catalog number: 69504), and long-range PCR reactions were performed using Q5 High-Fidelity 2× Master Mix (New England Biolabs, Ipswich, WA, USA, catalog number: M0492). An amount of 100 ng genomic DNAs were used as the template, and the PCR conditions were: first, 1 cycle of 98 °C for 30 s, followed by 40 cycles of 98 °C for 10 s, 66 °C for 30 s, and 72 °C for 2 min. The PCR products were subjected to 1% agarose gel electrophoresis and the DNA bands of interest were purified using QIAquick Gel Extraction Kit (Qiagen, catalog number: 28706) and subjected to direct Sanger sequencing (Genewiz) and analyzed using FinchTV (Geospiza). Specifically, for NAB2exon6-STAT6exon17 fusion allele, the forward primer (5′-GGTCATGTCCAAGGCTGACGCCGCCCCCTG-3′) is located within exon 6 of NAB2, and reverse primer (5′-GTAGCTGGGACATAACCCCTGCCATCCTTACC-3′) is located within exon 17 of STAT6. For wild type NAB2 allele, the forward primer (5′-GGTCATGTCCAAGGCTGACGCCGCCCCCTG-3′) is in exon 6 of NAB2 gene; and the reverse primer (5′-CCTCCCCTCCCTGGCTGTGCGTAGCTCTGT-3′) is in intron 6 of NAB2 gene.

For RT-PCR, total RNAs were harvested using RNeasy Mini kit (Qiagen, catalog number: 74106) and cDNAs were made using QuantiTect Reverse Transcription kit (500 ng RNA, Qiagen, catalog number: 205311). The cDNAs were then subjected to PCR reactions using Q5 High-Fidelity 2× Master Mix and the PCR conditions were: first, 1 cycle of 98 °C for 30 s, followed by 40 cycles of 98 °C for 10 s, 63 °C for 30 s, and 72 °C for 1 min. Specifically, for the NAB2exon6-STAT6exon17 fusion transcript, the forward primer (5′-CCTGTCTGGGGAGAGTCTGGATG-3′) is in exon 5 of NAB2 gene, and the reverse primer (5′-GGGGGGATGGAGTGAGAGTGTG-3′) is in exon 20 of STAT6 gene. The PCR products were subjected to 1% agarose gel electrophoresis and the DNA bands of interest were purified using QIAquick Gel Extraction Kit and subjected to direct Sanger sequencing (Genewiz) and analyzed using FinchTV (Geospiza).

### 2.3. Western Blot

To prepare whole cell lysates, cell pellets were washed with ice-cold PBS and then resuspended in RIPA buffer (Cell Signaling, catalog number: 9806) with protease/phosphatase inhibitors (Cell Signaling, catalog number: 5872). The resuspended cells were homogenized 10 times using a 1 mL syringe with 30 Gauge needles (VWR, catalog number: 328411), followed by 30 min incubation on ice. The cell suspensions were then centrifugated at 16,000× *g* rpm for 10 min and the supernatants were collected, and the protein concentrations were determined using Pierce BCA Protein Assay kit (Thermo Fisher Scientific, catalog number: 23227). For Western blot, 30 µg protein were used with anti-STAT6 C-terminus primary antibody (1:1000, Abcam, Cambridge, UK, catalog number: ab32520) and mouse anti-rabbit horseradish peroxidase-conjugated secondary antibody (1:5000, Santa Cruz Biotechnology, catalog number: sc-2357). The signal was visualized using the ChemiDoc XRS+ imaging system (BIO-RAD, Hercules, CA, USA, catalog number: 1708265) The reversible Ponceau staining was used as the alternative loading control [33,34].

### 2.4. Cell Proliferation Assay

80,000 HCT116 or NS-poly cells were seeded onto a 12-well plate (Greiner Bio-One, Monroe, NC, USA, catalog number: 665180) without hygromycin. At 24, 48 and 72 h, cells were trypsinized with 250 µL 0.25% trypsin-EDTA at 37 °C for 5 min. Trypsin-EDTA was then neutralized by adding 750 µL of complete medium. The cell suspension was then counted using a hemacytometer (Sigma-Aldrich, St. Louis, MO, USA, catalog number: Z359629). All experiments were performed in triplicates.

### 2.5. Wound Healing Assay

400,000 HCT116 or NS-poly cells were seeded onto a 6-well plate (Greiner Bio-One, catalog number: 657160) without hygromycin. After 48 h, the growth media was removed, and the confluent cells were scratched from top to bottom of the well using a 20 µL pipet tip. The cells were then gently washed with PBS twice to remove any cell debris. Finally, 2 mL of DMEM media supplemented with 0.1 mM MEM non-essential amino acids, 0.045 units/mL of Penicillin, and 0.045 units/mL of Streptomycin was added. The brightfield images (magnification 10×) were then captured every 2 h (up to 48 h) using an Olympus IX81 microscope (Tokyo, Japan). Data collection and processing were performed in the software package Slidebook 5.0.

### 2.6. RNA Sequencing (RNA-Seq)

For RNA-seq, total RNAs were harvested using RNeasy Mini kit and sample purities were evaluated using OD260/OD280 (1.8–2.2). The Genewiz Standard RNA-Seq service was employed, which requires 2 µg of total RNAs in 10 µL nuclease-free water. The assays were performed on an Illumina HiSeq platform (2 × 150 bp configuration, single index) and the outputs contained ~50 million reads per sample. All samples had mean quality scores larger than 30, and the summary sequencing statistics were presented in Appendix A. For data analysis, the human reference genome (UCSC hg38, https://genome-idx.s3.amazonaws.com/hisat/hg38_tran.tar.gz (accessed on 1 September 2021)) was used, and a pipeline consisting of HISAT2, StringTie and DESeq2 was employed to identify differentially expressed genes between the wild type HCT116 and NS-poly cells.

### 2.7. ASO Treatment

All ASOs were dissolved in PBS. ASO treatment was performed either with transfection reagents (RNAiMAX, Thermo Fisher Scientific, catalog number: 13778975) or without transfection reagents (free delivery by directly adding to the growth media). Cells were seeded in complete growth medium. After 18 h, ASOs were delivered using RNAiMAX following the manufacturer’s recommendation or added directly into the growth medium. After 48 h, the cells were harvested for further analysis.

### 2.8. Real-Time RT-PCR (Reverse Transcription-PCR)

For real-time RT-PCR assays, total RNAs were extracted using RNeasy Mini Kit (Qiagen, catalog number: 74106). First-strand cDNAs were synthesized using QuantiTect Reverse Transcription kit (500 ng RNA, Qiagen, catalog number: 205311). Next, quantitative PCR was performed using the KAPA SYBR FAST universal qPCR Kit (Kapa Biosystems, Wilmington, MA, USA, catalog number: KK4601), with GAPDH as the internal control. The forward primer (P25) for GAPDH was: 5′-AATCCCATCACCATCTTCCA-3′ and the reverse primer (P26) for GAPDH was: 5′-TGGACTCCACGACGTACTCA-3′. Quantitative analysis was performed using the 2^−ΔΔCt^ method. Fold-change values were reported as means with standard deviations.

### 2.9. Recombinant AAV2 Viral Vector Production

For primary AAV2 viral vector production, the AAV-2 Helper Free System (Agilent, catalog number: 240071) was used. Briefly, AAV-293 cells were seeded at 70–80% confluency. The cells were then transfected with pAAV-RC, pHelper and pAAV expression plasmid using JetPRIME (Polyplus Transfection, catalog number: 101000046). The cells were harvested after 72 h and subjected to four rounds of freeze/thaw cycles using a dry ice-ethanol bath and a 37 °C water bath. The cells were then centrifuged at 16,000 g for 10 min at room temperature, and the supernatant was transferred to new Eppendorf tubes stored at −80 °C. The physical titers of primary AAV2 viral vectors were determined by qPCR AAV Titer Kit (Applied Biological Materials, Richmond, BC, Canada, catalog number: G931).

### 2.10. Fluorescence Microscopy

Microscopy was performed 48 h post-transfection. The live cells were grown on 12-well plates (Greiner Bio-One) in the complete medium. Cells were imaged using an Olympus IX81 microscope (Tokyo, Japan) in a precision-controlled environmental chamber. The images were captured using a Hamamatsu ORCA-03 cooled monochrome digital camera (Hamamatsu, Japan). The filter sets (Chroma Technology, Bellows Falls, VT, USA) are as follows: ET500/20× (excitation) and ET535/30 m (emission) for Yellow Fluorescent Protein (YFP). Data collection and processing was performed in the software package Slidebook 5.0. All images within a given experimental set were collected with the same exposure times and underwent identical processing.

### 2.11. Flow Cytometry

A total of 48 h post-transfection, cells from each well of the 12-well plates were trypsinized with 250 µL 0.25% trypsin-EDTA at 37 °C for 5 min. Trypsin-EDTA was then neutralized by adding 750 µL of complete medium. The cell suspension was centrifuged at 1000 rotations per minute for 5 min, and after removal of supernatants, the cell pellets were resuspended in 0.5 mL phosphate-buffered saline buffer. The cells were analyzed on a BD Biosciences LSRFortessa flow analyzer (San Jose, CA, USA). YFP was measured with a 488 nm laser, a 535 nm emission filter and a 545/35 band-pass filter. For data analysis, 100,000 events were collected. A forward scatter/side scatter gate was generated using an un-transfected negative sample and applied to all cell samples. All experiments were performed in triplicates.

### 2.12. Mouse Xenograft

Foxn1^nu^ athymic nude mice were purchased from The Jackson Laboratory (Bar Harbor, ME, USA, catalog number: 002019) and maintained in a pathogen-free facility, in accordance with Protocol #21-05 approved by IACUC of University of Texas at Dallas. For mouse xenograft, 5 million NS-poly cells were resuspended in 100 µL of PBS and injected subcutaneously into the right flank region of 4–6-week-old female Foxn1^nu^ athymic mice under anesthesia using isoflurane (Covetrus, Portland, ME, USA, catalog number: 11695-6777-2). The tumor size was measured using a digital caliper and the tumor volume was calculated as (L × W × W)/2, where L was tumor length and W the tumor width. Mice were monitored daily for general health (signs of dehydration, cachexia, weight loss, and dyspnea) and euthanized when tumors reached 2 cm^3^ in volume or mouse body weight decreased by more than 20%.

### 2.13. H and E (Hematoxylin and Eosin) Staining

The paraffin-embedded tissue slides (10 µm) were first deparaffinized by heating at 60 °C for 10 min. The slides were then re-hydrated and staining using hematoxylin 560 (Leica, Wetzlar, Germany, catalog number: 3801570). The slides were then counterstained using Eosin Y 515 (Leica, catalog number: 3801615). After dehydration, one drop of mounting medium (Abcam, catalog number: ab64230) and a glass cover were added to each slide, and slides were then observed using a Leica DMi1 Microscopy.

### 2.14. CT Scan

Each mouse underwent micro-CT (OI/CT, MILabs, Utrecht, The Netherlands) imaging using an accurate, ultra-focus image scan at a step angle of 0.250 degrees at 1 projection per step and a binning size of 1. The micro-CT tube settings were set at a voltage of 50 kV, a current of 0.21 mA, and an exposure time of 75 ms. Images were converted to DICOMs using vendor software OsiriX (version 12.0.0).

## 3. Results

### 3.1. Generation of NAB2exon6–STAT6exon17 Gene Fusion Cell Models Using Genome Editing

A recent breakthrough in understanding SFTs was the identification of recurrent NAB2–STAT6 gene fusions in almost all SFT tissue samples [4,5]. Specifically, the breakpoints within NAB2 and STAT6 genes, which are adjacent on chromosome 12q13, induce the inversion of DNA fragments and subsequently the expression of NAB2–STAT6 transactivators.

Thus far, seven distinct NAB2–STAT6 gene fusion types (NAB2exon2–STAT6exon5, NAB2exon4–STAT6exon2, NAB2exon4–STAT6exon4, NAB2exon5–STAT6exon16, NAB2exon6–STAT6exon16, NAB2exon6–STAT6exon17, and NAB2exon7–STAT6exon2) have been discovered to commonly account for pathologic variation and tumor aggressiveness in SFTs. Herein all exons and introns of the human STAT6 gene were numbered in accordance with the latest NCBI reference transcript (NM_001178078.2).

We first aimed to use the CRISPR/spCas9 system to generate an HCT116-based NAB2exon6–STAT6exon17 fusion cell model. We noted that for this fusion type, the inverted sequence was approximately 5 kb (Figure 1a, from NAB2 intron 6 to STAT6 intron 16), and thus within the size limit for a SpCas9-mediated knock-in strategy (typically less than 10 kb) [35,36,37,38,39,40]. Here, the human colorectal cell line HCT116 was employed due to its high transfection and editing efficiency, as compared to SFTs’ (debated) cell of origin of mesenchymal cells [41]. Our engineered cell lines did indeed recapitulate pathogenic tumor aspects, allowed us to profile the impact of the fusion to their transcriptome, and assess the efficacy of RNA-targeting methodologies in vitro and in vivo.

Two sgRNAs were designed to target intron 6 of NAB2 gene (5′-CAGAAATTCCAGCGCAACCGAGG-3′, PAM underlined) and intron 16 of STAT6 gene (5′-GGAGGAAGTGGGTGACAGGAAGG-3′, PAM underlined), respectively. Next, a homologous recombination (HR) repair template, which contains the inverted sequence of original NAB2 intron 6-STAT6 intron 16, was designed (Figure 1a). In addition, a hygromycin resistance gene cassette, which was flanked by two FRT (flippase recognition target) sites, was placed after the exon 7 of NAB2. HCT116 cells were transiently transfected with PCMV–SpCas9, the two sgRNA constructs, along with the HR repair template, and after 48 h, treated with 200 µg/mL hygromycin for two weeks. The established polyclonal stable cell line (named as NS-poly) was further transfected with a flippase-expressing construct to remove the hygromycin resistance gene cassette. Finally, the transfected cells were sorted into single cells using flow cytometry cell sorter, and three monoclonal stable cell lines were established (named as NS-11, NS-17, and NS-23).

To determine the genotype of our stable cells, we first harvested genomic DNAs using DNeasy Blood & Tissue kit and performed genotyping PCR reactions using primers P1 and P2 (Appendix A) for the NAB2exon6–STAT6exon17 fusion allele. As shown in Figure 1b, the expected amplicon (1194 bp) was observed in NS-poly, NS-11, NS-17, and NS-23, but not in HCT116 wild type cells, indicating that all four stable cell lines contained the fusion allele. Furthermore, we subjected the genomic DNAs to PCR reactions using primers P3 and P4, and the expected amplicon (1020 bp) was observed in all stable cells as well as in the wild-type sample. Taken together, these results showed that all four stable cell lines are heterozygous.

We further confirmed successful integration of the NAB2–STAT6 gene fusion at the RNA level. We extracted total RNAs from the stable cells using RNeasy kit and subjected the RNA samples to RT-PCR for the fusion transcript (441 bp, forward primer P5: 5′-CCTGTCTGGGGAGAGTCTGGATG-3′, exon 5 of NAB2 gene; reverse primer P6: 5′-GGGGGGATGGAGTGAGAGTGTG-3′, exon 20 of STAT6 gene). Like our genomic PCR results, all four stable cell lines yielded the expected band, which was absent in the original HCT116 cells (Figure 1b). Subsequent Sanger sequencing further confirmed the fusion type as NAB2exon6–STAT6exon17 (Figure 1c, breakpoint adjacent region between NAB2 exon 6: 5′-CCTCTCGCAG-3′ and STAT6 exon 17: 5′-CTGAACAGAT-3′ highlighted in white). Lastly, we prepared whole cell lysates using RIPA buffer, which were subsequently subjected to Western blot using a STAT6 C-terminus-targeting antibody. As shown in Figure 1d and Appendix A, all four fusion stable cells expressed the NAB2exon6–STAT6exon17 fusion protein (expected size: 76 kD), which was not observed in the HCT116 wild type cells.

For in vitro characterization of NS-poly and its parental HCT116 cell line (HCT116), we first performed cell proliferation assay using a hemacytometer, which showed no significant difference of cell growth rates between the two cell lines (Appendix A). Next, the wound healing assays were used to measure the cell migratory potentials (Appendix A, for wild type HCT116 at 0 and 48 h), which showed that the NS-poly cells had a higher motility rate compared to HCT116 cells (Appendix A). As an example (Appendix A), 24 h after scratching, the NS-poly cells closed the wound by 283.5 pixels, compared to 202.8 pixels for wild type HCT116 cells.

### 3.2. RNA-Sequencing (RNA-Seq) Analysis of NAB2exon6–STAT6exon17 Gene Fusion Cell Models

NAB2–STAT6 fusions have been shown to function as transcriptional activators and upregulate the expression of cancer-promoting EGR1 target genes including FGFR1 and NTRK1, as well as IGF genes in SFT patient samples [4,5]. To systemically characterize the pathway and network perturbations induced by NAB2–STAT6 fusions, we subjected our HCT116-based NAB2–STAT6 fusion stable cells to RNA-sequencing (RNA-seq). Briefly, total RNAs were harvested from HCT116 wild type (3 replicates) as well as NS-11, NS-17, and NS-23 cells (one sample for each monoclonal cell line), and subsequently RNA-seq assays were performed on an Illumina HiSeq platform (Genewiz). Next, an analysis pipeline consisting of HISAT2, StringTie and DESeq2 was used to identify differentially expressed genes between the wild type and fusion cells [42]. As shown in Figure 2a (adjusted *p*-values < 0.01 using FDR/Benjamini–Hochberg multiple testing correction) and Figure 2b, the expression of NAB2–STAT6 fusion induced extensive changes at the transcriptional levels (Appendix A, 198 differentially expressed genes using relatively stringent filtering conditions: adjusted *p*-values < 0.01, and |log_2_(fold-change)| > 3). Subsequently, we performed signaling pathway analysis using the PANTHER classification system and identified candidate genes which contained both MSTRG numbers and corresponding gene names (131 genes, Appendix A) [43]. As shown in Figure 2c and Appendix A, multiple cancer-related signaling pathways have been identified, including the FGF signaling pathway, VEGF signaling pathway, EGF receptor signaling pathway, and Ras pathway. Additionally, the identified angiogenesis pathway is consistent with previous reports of the interfacing between STAT6 and neoangiogenesis [44]. We emphasize that analyses of these perturbed genes and signaling pathways could provide crucial insights into the development of SFT cancers, as well as allow rational design of targeted therapeutic options. Specifically, we observed that a particular glycosyltransferase family member, MGAT5 (Mannoside Acetylglucosaminyltransferase 5), was overexpressed in all three fusion monoclonal cell lines, but absent in wild type control replicates (Figure 2b), which implied that protein glycosylation may play a role in SFT pathology, if similar results can be observed in additional engineered or primary SFT cell models.

### 3.3. In Vitro Targeting of NAB2–STAT6 Fusion Transcripts Using Fusion-Specific Antisense Oligonucleotides (ASOs)

Unlike the noncancerous cells, in SFT cancer cells the fusion of NAB2 and STAT6 transcripts create novel sequences at the junction site (e.g., 5′-CCTCTCGCAG|CTGAACAGAT-3′ for NAB2exon6–STAT6exon17 fusion type, | denotes the breakpoint), which could serve as the basis for fusion-specific RNA targeting. Accordingly, for NS-poly cells, we designed three fusion-specific ASOs with 2′-O-methoxyethyl modifications (fusion2, fusion4, and fusion6, Appendix A). Subsequently, we tested the in vitro targeting efficacies against the NAB2–STAT6 fusion transcripts (NAB2–STAT6 fusion-specific primers, P21 and P22, Appendix A) using both liposome-based (Lipofectamine RNAiMAX, Invitrogen) and free delivery methods.

A control ASO (Integrated DNA Technologies, 5′-C*G*T*T*A*A*T*C*G*C*G*T*A*T*A*A*T*A*C*G*-3′, * denotes phosphorothioate modification), designed not to target any human RNA transcripts, was also included. As shown in Figure 3a, using the RNAiMAX delivery (1 µM final concentration due to possible non-specific cytotoxicity effects of ASOs at higher concentrations [45,46,47]), all three candidate ASOs suppressed the expression of NAB2–STAT6 fusion transcript, with Fusion6 ASO yielded the highest efficacy (58% suppression, *p*-value < 0.05). In contrast, none of the three ASOs induced any significant suppression using the free delivery method (Figure 3b). These results implied that our cell line model, which is colonic epithelial HCT116-based, may not be compatible with the free uptake mechanism.

Next, NS-poly cells were treated with Fusion6 ASO using RNAiMAX at different dosages (0 nm, 10 nm, 30 nm, 100 nm, 300 nm, and 1 µM). As shown in Figure 3c, while Fusion6 ASO efficiently suppressed the expression of NAB2–STAT6 transcript (primers P21 and P22, IC50: 356.6 nM), no suppression of wild type STAT6 transcript (wild type STAT6-specific primers P23 and P24, Appendix A) was observed. Lastly, we transfected NS-poly cells with either Fusion6 or the control ASO at 1 µM concentration. As shown in Figure 3d, Fusion6 ASO significantly suppressed the proliferation of NS-poly cells after 72 h (22.1% suppression, *p*-value < 0.05). Our results indicated that Fusion6 ASO-based RNA targeting can affect both RNA expression and cell proliferation rates in NS-poly cells.

### 3.4. In Vitro Targeting of NAB2–STAT6 Fusion Transcripts Using AAV2-Mediated Fusion-Specific CRISPR/CasRx

The *Rfx*Cas13d (CasRx)-based RNA editing has been reported to be highly efficient in mammalian cells [30,31,32]. Additionally, the presence of a PAM (protospacer adjacent motif) is not required for Cas binding. Thus, for our NAB2 exon6–STAT6 exon17 fusion type, a 30–nt fusion mRNA junction-targeting pre-gRNA sequence was designed (5′-TACCCATCTGTTCAG|CTGCGAGAGGTGGCT-3′, | denotes the break point), and subsequently cloned into a CMV-CasRx-U6-sgRNA.NAB2STAT6 construct (Figure 4a). We first evaluated its editing efficacies using a YFP (Yellow Fluorescent Protein) reporter construct, which contained the corresponding target site after the start codon ATG. As shown in Figure 4b,c, upon delivery into HCT116 cells, this CasRx–pre-sgRNA complex potently suppressed the expression of YFP by 97% (*p*-value < 0.001), compared to a negative control (NC) sgRNA which targets the human *AAVS1* locus (5′-ACCCAGAACCAGAGCCACATTAACCGGCCC-3′).

Next, we noted that CasRx (930 aa) is the smallest discovered member in the Cas13 protein family, which makes it fully compatible with most AAV packaging systems. Thus, we prepared the NAB2 exon6-STAT6 exon17 fusion-targeting CasRx AAV2 viral vectors, and transduced NS-poly cells at various MOIs (Multiplicity of Infection). As shown in Figure 4d, the viral vectors suppressed the expression of the NAB2–STAT6 fusion transcript in a dose-dependent manner (25% suppression at MOI 1000, *p*-value < 0.01, and 59% suppression at MOI 5000, *p*-value < 0.01). In contrast, the vectors exerted no statistically significant suppression effects on the wild type STAT6 transcript at lower MOIs (Figure 4e, *p*-value > 0.20 at MOI 1000). It should be noted that a mild suppression of STAT6 transcript was observed at higher MOIs (19% suppression at MOI 5000, *p*-value < 0.01) which may indicate AAV2-associated cell toxicity [48]. Take together, these results demonstrated that our AAV2-mediated CasRx system was both effective and specific on the fusion transcript at lower MOIs, and the MOI of 1000 was used for the subsequent ex vivo targeting experiments.

### 3.5. Ex Vivo Targeting of NAB2–STAT6 Fusion Transcripts Using AAV2-Mediated Fusion-Specific CRISPR/CasRx System

To evaluate whether AAV2-mediated NAB2–STAT6 fusion-targeting CasRx system could exert in vivo therapeutic benefits for SFTs, we first created a subcutaneous (subQ) NS-poly xenograft model using 5 million NS-poly cells and Foxn1^nu^ athymic nude mice (The Jackson Laboratory). As an example, 5 weeks post-injection, one xenograft measured ~2.3 cm^3^, which was then harvested for preparation of paraffin sections (10 µm). Subsequent Hematoxylin and eosin (H and E) staining showed both necrotic core and neovascular blood vessels within the mass (Figure 5a).

For ex vivo evaluation, NS-poly cells were transduced with NAB2–STAT6 fusion-targeting CasRx AAV2 viral vectors (NAB2-STAT6 CasRx) at MOI of 1000 in a 10 cm Petri dish. After 48 h, treated cells were harvested for xenograft and tumor sizes were measured twice a week using a digital caliper. PBS-treated NS-poly cells were used as the negative control. As shown in Figure 5b, 29 days post-implantation, tumor sizes were significantly smaller in the NAB2–STAT6 CasRx-treated mice (1340.7 mm^3^ compared to 2963.4 mm^3^ for PBS-treated mice, *p*-value < 0.05), which were corroborated by high-resolution computed tomography (CT) scans with axial and sagittal views of the tumors prior to excision (Figure 5c). Taken together, our data demonstrated the potential therapeutic benefits of AAV2-mediated fusion-specific CRISPR/CasRx system in suppressing the SFT tumor growth.

## 4. Discussion

A major bottleneck to targeted therapy development is the lack of cell models of SFT. Using the CRISPR genome editing technologies, we have built several in vitro SFT cell line models (e.g., NS-poly) in HCT116 cells, partially due to their high transfection and genome editing efficiencies. Having the HCT116 cells with and without the NAB2–STAT6 gene fusion is crucial to isolating the impact of the fusion’s contribution alone on the pathologic variation and tumor aggressiveness. These cell lines better resemble primary SFTs compared to cell models used in previous studies. As an example, a NAB2–STAT6 fusion cDNA construct was stably integrated into mouse fibroblast cell line NIH-3T3 [7]. The resulting cells, although partially recovering characteristics of SFTs, do not preserve important genetic information of the original NAB2–STAT6 due to the lack of sequences including endogenous NAB2 promoters, 5′-UTRs of NAB2, and 3′-UTRs of STAT6. In contrast, our CRISPR-generated cell models preserved all original NAB2–STAT6 gene fusion information.

Our results here highlight the oncogenic role of the NAB2–STAT6 in SFT. Namely, compared to the wild type HCT116 cells, the presence of the NAB2–STAT6 fusion led to an increase in cell migration. These results suggest the downstream pathways of the gene fusion may influence cell motility. RNA-sequencing results indicate cell motility and growth signaling pathways are affected by the NAB2–STAT6 fusion (e.g., RAS, EGFR, VEGF, FGF, etc.). Encouragingly, fusion specific ASO treatment reduced the fusion transcript by 58% and cell proliferation by 22% after 72 h in vitro. Similarly, AAV2-mediated CRISPR/CasRx treatments reduced the fusion transcript by 59% in vitro and tumor growth by 55% after 29 days ex vivo. Notably, the parent wild type HCT116 cells are inherently oncogenic. Therefore, the fact that an observable reduction in the growth of the HCT116 cells with the NAB2–STAT6 gene was obtained by exclusively reversing the NAB2–STAT6 expression, further emphasizes the role of the NAB2–STAT6 gene fusion in tumorigenicity.

In this study, we have designed ASOs which specifically target the NAB2–STAT6 fusion transcripts but not wild type NAB2 or STAT6 transcripts (Figure 3). It should be noted that this design strategy depends on specific NAB2–STAT6 fusion types (NAB2exon6-STAT6exon17 type in our case), which increases the specificity and reduces adverse off-target side-effects. However, the cost of a personalized approach may limit its broad clinical use. There have been shown to be at least six distinct fusion types that may account for pathologic variation and tumor aggressiveness seen in SFTs [6]. Alternatively, we note that all known NAB2–STAT6 fusions contain the C-terminal of the human STAT6 transcript, which includes the 3′-UTR (untranslated region) sequence (~1.1 kb). More importantly, previous studies have demonstrated that depletion of wild type STAT6 may elicit beneficial effects [44,49,50,51]. Therefore, novel ASOs could be designed to target the 3′-UTR of STAT6, which could in theory suppress the expression levels of both wild type STAT6 and all known NAB2–STAT6 fusion transcripts, especially if a SFT-tumor-specific ASO delivery method is developed.

Next, we note that for CRISPR/CasRx systems, there are discrepancies in the current literature with respect to their specificities, or the collateral degradation of non-target RNAs [52,53]. As an example, You and colleagues showed that the collateral activities of the CasRx system positively correlate with the abundance of target RNA, which could subsequently induce the cleavage of 28 s rRNA and cell toxicity. Furthermore, both CasRx-reactive antibodies and CasRx-responding T cells have been reported in healthy human donors [54]. Although not observed in our CasRx experiments, RNA editing may elicit cytotoxicity. Taken together, thorough biosafety studies of CasRx are required before being applied to clinical treatments.

Finally, we do want to highlight the limitations of our current study. Firstly, although our engineered cells provide a means to differentially isolate the effects of the fusion, our choice of cell line (HCT116) may not be ideal. SFTs are believed to originate from mesenchymal stem cells with fibroblastic differentiation, but not from colonic epithelial cells. Accordingly, although our RNA-seq assay and the subsequent PANTHER signaling pathway analysis points to the potential involvement of VEGF and EFGR signaling, their clinical significance in SFT remains unclear, as both RAS and TP53 genes are reportedly mutated in HCT116 and both genes are known to have interactions with VEGF or EFGR signaling [55,56]. Independent of the cell type of origin, primary occurrences are observed throughout the body with various presentations. For example, Bieg M. et al. noted that pleuropulmonary SFTs are less cellular and more collagenous, whereas retroperitoneum, pelvic, and meningeal SFTs have more ovoid or round cell morphologies [8]. Thus, the slight uncertainty in presentation and cell type of origin, renders the study of the fusion behavior in different cellular backgrounds still insightful. Notably, although not necessarily derived from cholinergic cells, SFTs have been reportedly present as a polyp in the colon tissue [57].

In follow up studies, we will engineer all the NAB2–STAT6 fusions in mesenchymal fibroblastic cell lines, and subsequently re-examine the targeting effects of our candidate ASOs and the CRISPR/CasRx system in these engineered cell models and SFT patient derived primary cell lines (e.g., SFT-T1 and SFT-T2 cell lines from Ghanim et al. [58]). Similarly, we will confirm our signaling pathway analysis results in these new cell models using RNA-seq assays, and upon confirmation, perform additional immunohistochemistry (IHC) or immunofluorescence (IF) assays for protein targets in such signaling (e.g., PI3K for VEGF signaling [59]) using the corresponding ex vivo tumor tissues.

Additionally, it should be emphasized that the use of ASOs remains challenging in a clinical setting, especially in terms of the delivery of the drug to the desired site (e.g., solid tumor tissues). Briefly, ASO drugs need to travel through the blood stream, pass through biological barriers, and withstand lysosomal degradation upon internalization by target cells. To address these difficulties, different chemical modification methods, such as the phosphorothioate (PS) backbone, 2′-MOE, and locked nucleic acids (LNA) have been developed to increase their stability. Additionally, various delivery vehicles, from DNA nanostructures to exosomes, have been used to increase the delivery efficiency.

Lastly, the ex vivo efficacy of our NAB2–STAT6 fusion-targeting CRISPR/CasRx system was evaluated using a xenograft model, which may not fully capture the pathological demonstrations of SFT. Thus, in future studies, we plan to reassess candidate RNA-based therapeutics (ASOs and CRISPR/CasRx) using our recently developed SFT PDX (patient-derived xenograft) model [60], and additionally explore the combinations of ASOs and CRISPR/CasRx for their potential synergistic therapeutic effects.

## 5. Conclusions

Our study demonstrated the potential of using RNA therapeutics (antisense oligonucleotides and the CRISPR/CasRx system) to target the pathological NAB2–STAT6 fusion transcripts in SFTs. Further investigations are needed to evaluate their in vivo efficacies and safety before being translated into clinical practice.

## Figures and Tables

**Figure 1 cancers-15-03127-f001:**
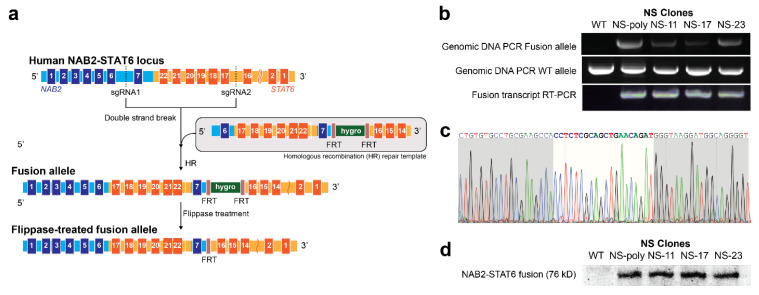
Preparation of HCT116-based NAB2exon6–STAT6exon17 fusion stable cell lines using the CRISPR/SpCas9 system. (**a**) Schematic illustration of the CRISPR/SpCas9 homologous recombination process to create NAB2exon6–STAT6exon17 fusion type in HCT116 cells. (**b**) Genotyping of NAB2exon6–STAT6exon17 fusion stable cells. Genomic DNA PCR-and RNA RT-PCR assays confirmed that the fusion stable cells were heterozygous. (**c**) Sanger sequencing result for RT-PCR amplicons confirmed the fusion type. (**d**) Western blot confirmed the expression of NAB2exon6–STAT6exon17 fusion proteins in the fusion stable cells.

**Figure 2 cancers-15-03127-f002:**
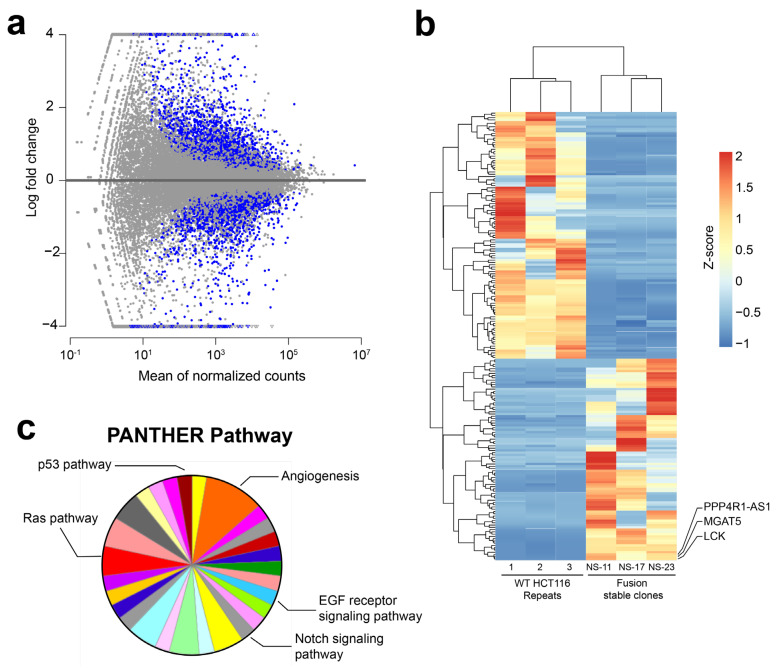
Transcriptome analysis of monoclonal NAB2exon6–STAT6exon17 fusion cells with their control HCT116 cells. (**a**) MA-plot analysis of monoclonal fusion cells vs. control HCT116 cells displaying differentially expressed (blue) genes (adjusted *p*-values < 0.01). (**b**) Heatmap showing differentially expressed genes between fusion stable cells and control HCT116. Expressions of MGAT5 were upregulated in three monoclonal fusion cells. (**c**) PANTHER analysis showed that multiple cancer-related signaling pathways were affected in monoclonal fusion cells.

**Figure 3 cancers-15-03127-f003:**
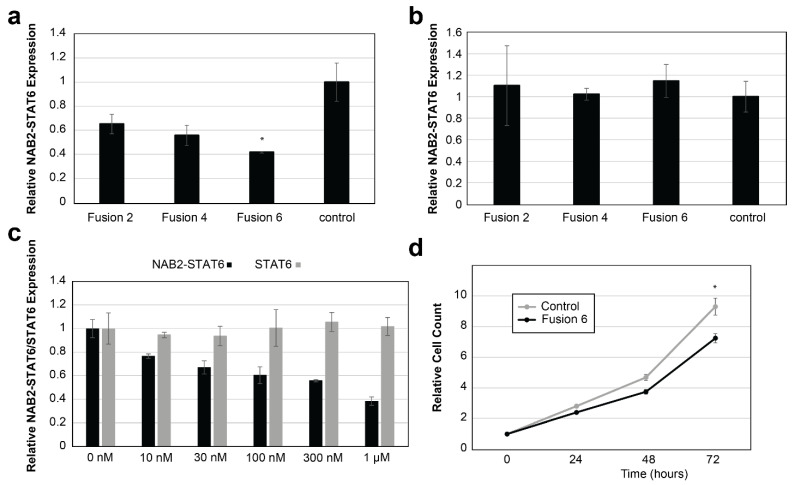
In vitro targeting of NAB2–STAT6 fusion transcripts using fusion-specific ASOs. (**a**) In vitro targeting of NAB2–STAT6 fusion transcripts by ASOs using RNAiMAX-mediated delivery. Fusion6 ASO suppressed the expression of NAB2–STAT6 transcript by 58%. (**b**) In vitro targeting of NAB2–STAT6 fusion transcripts by ASOs using gymnotic delivery. (**c**) Fusion6 ASO efficiently and specifically suppressed the expression of NAB2–STAT6 transcript. (**d**) Fusion6 ASO suppressed the proliferation of NS-poly cells compared to the negative control ASO. (* indicates *p*-value < 0.05 using two-tailed *t*-test).

**Figure 4 cancers-15-03127-f004:**
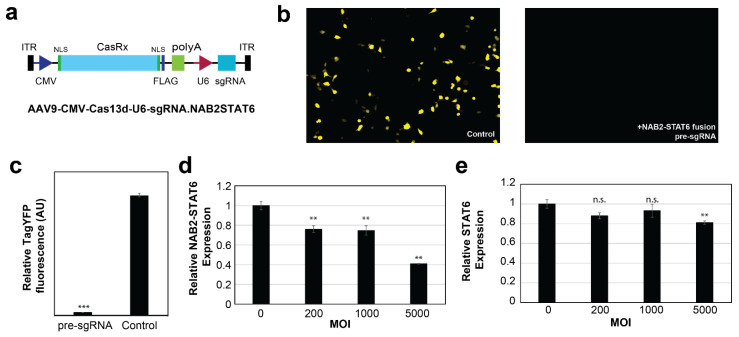
In vitro targeting of NAB2–STAT6 fusion transcripts using fusion-specific CRISPR–CasRx system. (**a**) Schematic illustration of the AAV2-based CRISPR/CasRx construct. (**b**) Fluorescence microscopy and (**c**) flow cytometry assay showed that the CRISPR/CasRx efficiently suppressed YFP expression, whose transcript contains a CasRx target site after the start codon (ATG). (**d**) Quantitative RT-PCR showed that AAV2-based NAB2–STAT6 fusion targeting viral vectors suppressed the expression of NAB2–STAT6 fusion transcripts in NS-poly cells. (**e**) AAV2-based NAB2–STAT6 fusion targeting viral vectors only mildly suppressed the expression of wild type STAT6 fusion transcript in NS-poly cells at the highest tested MOI (19% at MOI 5000). Two-tailed *t*-tests were used for all statistical testing. *** indicates *p*-value < 0.001, ** indicated *p*-value < 0.01, and n.s. indicates *p*-value > 0.05.

**Figure 5 cancers-15-03127-f005:**
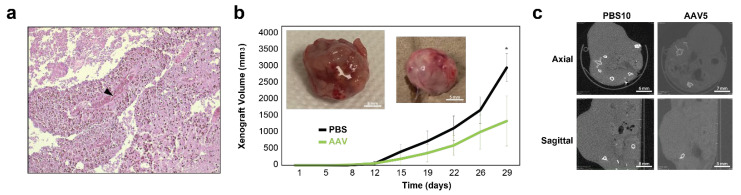
Ex vivo targeting of NAB2–STAT6 fusion transcripts using fusion-specific CRISPR–CasRx system. (**a**) Hematoxylin and eosin (H and E) staining of NS-polys cells-derived mouse xenograft. The arrow indicates the formation of neovascular blood vessels in the xenograft tissue. (**b**) Tumor growth curves of PBS-(black) or NAB2-STAT6 CasRx (green)-treated NS-poly cells subcutaneously injected in Foxn1^nu^ athymic nude mice. From 29 days post-implantation, NAB2–STAT6 CasRx-treated group showed slower tumor growth (* indicates *p*-value < 0.05 using two-tailed *t*-test). (inlet) Representative images of xenograft harvested from each treatment group; PBS-treated (**left**), NAB2-STAT6 CasRx-treated (**right**). (**c**) Representative high resolution CT scan images of mouse cross sections around the tumor; PBS-treated (**left**), NAB2–STAT6 CasRx-treated (**right**).

## Data Availability

Data collected for this article are available in the Appendix A.

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
