# Peer review of "Reduction of Tumor Growth with RNA-Targeting Treatment of the NAB2–STAT6 Fusion Transcript in Solitary Fibrous Tumor Models"

_cancers, 2023, doi:10.3390/cancers15123127_

Round 1

Reviewer 1 Report

The authors of this article have tried to use RNA targeting technology and CRISPR/CasRx to suppress expressions of NAB2-STAT6 fusion transcripts and reduce cell proliferation and tumor growth. The authors engineered NAB2-STAT6 fusion SFT cell models to evaluate the targeting efficiencies of ASOs and CRISPR/CasRx. They conclude that this has a potential as a novel therapeutic strategy for SFTs. They did a great work in showing that these systems can successfully reduce cell proliferation and tumor growth and that it has potential as therapeutic strategy. However, there are few things that the author can address to improve this article.

Line 314-317: The supplementary figure numbers are not correct. It should say supplementary 3a and 3b instead of 2a and 2b.

Line 308: Figure 1d (Western Blot). I would suggest the authors to provide the original western blot images and to have an internal loading control, for e.g., actin or GAPDH.

Line 314-315: Supplementary figure 3a (wound healing assay)- The authors should include the scale bar on the wound healing assay images.

Figure 3C: Relative NAB2-STAT6/STAT6 expression- The expression was going down until the 1 μm fusion 6 ASO concentration. Did the author test higher concentration (2 μm,3 μm,4 μm…) to see if they see further decrease in relative NAB2-STAT6/STAT6 expression or the expression just plateaus after certain concentration?

To further strengthen this article, the author should test ASO and CRISPR/CasRx systems on more NAB2-STAT6 fusion cell models. Ghanim, Bahil, et al established two patient-derived cell models characterized as SFT by the NAB2-STAT6 gene fusion. It will be good idea to test ASO and CRISPR/CasRX system in these patient derived cell lines, to confirm their findings. Ghanim, Bahil, et al’s article also suggested two novel and potential treatment regimens for SFT’s. I was wondering if authors have ever thought of trying combination of ASO and CRSIPR/CasRX system with treatment regimens suggested by Ghanim, Bahil, et al to see if it has better potential of treating SFTs.           

Article link: Ghanim, Bahil, et al. "Trabectedin Is Active against Two Novel, Patient-Derived Solitary Fibrous Pleural Tumor Cell Lines and Synergizes with Ponatinib." Cancers 14.22 (2022): 5602.                                  

Reviewer 2 Report

Comments for manuscript: 2403579

The manuscript here describes the role of using RNA-based therapies for the treatment of Solitary fibrous tumors.  Authors show that by specifically targeting NAB2-STAT6 fusion using the 31 CRISPR/SpCas9 system can be used as potential therapy for the treatment of SFT's.

This study is novel and may be of interest as the understanding of SFT progress and their treatment strategies are scarce. The results after treatment of SFT genetically engineered cells in terms of inhibition of cell migration and tumor growth shown are very modest.

This study is of importance but cannot be accepted in the current state and might require a revision to address the following comments:

In general, the results section is very long and describes the method in very detail that could be removed from there and instead can be moved to materials and methods section.

1.       For fig. 2A, to get the list of DEGs the p-value cut-off use is 0.1, can author explain why this was chose and what is the gene expression profile when cut off < 0.05 was used?

2.       For fig. 4, all graphs lack statistical analysis and the claim that AAV2-based NAB2-STAT6 fusion targeting viral 421 vectors only mildly suppressed the expression of wild type STAT6 fusion transcript does not hold true without stats.

3.       From the ex-vivo tumors, IF or IHC should be performed to stain for protein downstream to VEGF and EGFR signaling.

4.       To further validate that this RNA based therapy can be effective PDX of SFT should be targeted with specific ASO’s and RNA-based targeted approach.

Reviewer 3 Report

In the manuscript by Li et al., the authors describe the establishment of NAB2-STA6 fusion using CRISPR/Cas technology for the generation of a model for solitary fibrous tumours. The authors used state of art techniques to introduce the pathognomonic NAB2-STAT6 fusion. Furthermore, they performed several tests to show the presence of fusion and the effect of the fusion on the tested cells in an isogenic model system. They also attempted to show the ASO effect on the proliferation of the herein-developed model system.

Technically the study is sound, however, it is far from being well justified in how the selection of the used model cell line was made. The use of a colon carcinoma cell line to study mesenchymal tumours is rather unusual and unhandy. Even though the cellular origin of the SFT is unknown, it is certain that not colonic epithelial and in general not epithelial compartment. For this reason, the authors should make a reasoning of the used system and also make a paragraph with the limitations of the used system on the conclusions in this study including applications. Similarly, the use of ASO is in principle a good system in cell line models but remains less feasible in real clinical settings, targeting the tumours, reaching the cell of interest, specificity of the delivery material and so on. These should be all addressed.

Other concerns with minor issues that should be addressed are listed here.

Do not use hemangiopericytoma see WHO book, p 104, as authors refer to the WHO book, please read the chapter and avoid the before-mentioned term.

2.13 is too detailed a description for a standard H&E staining. This can be shortened.

Line 256-258 even if the origin of SFT is debated mesenchymal, for sure not colorectal carcinoma derived, should be addressed, see above.

Line 278 (NS-poly) is used as POLY-ns in other places. The use of the generated cell line name is not consistent and therefore confusing. Make it consistent.

Figure 1d, reference control protein expression is lacking from this WB, in this was the conclusion on the expression level is not possible. It looks strongly overexposed blot, so the expression level is difficult to call from this as stated in lines 308-309

Line 330-331 why use a p-value of 0.1? What type of multiple testing correction was used?

Line 343-345 This can not be concluded as we do not know how the possible expression of this gene would be regulated in a non-colon cancer-derived model.

MOI is in the range of 5000!!! Is this good? This is an unusually high MOI, authors should explain why they used this high value. Was the actual loaded viral genome tested by qPCR?

Figure 5a the arrow does not indicate a neovascular blood vessel, as far as it can be judged from the image. It also, looks like there was a staining or some nuclear marker (perhaps STAT6?) and the morphology is far from the SFT and looks as if much necrosis or secretion was there revealing more of the colo-rectal cancer type.

Line 473 and others, RAS is mutated in this line, TP53 is mutated as well and p16 is inactivated. So it is questionable to what level the identified pathways are reliable and to what degree are related to SFT where RAS  mutation is not found. At the same time, TP53 is rather an indicator of aggressiveness.

In lines 483-495 before the clinical use, it should be possible to reach specific and highly effective tumour targeting in human situations. The 3’-UTR might induce extra side effects unless a tumour-specific delivery method is devised.

Supplementary Figure 1 is a ponceau staining,  no way the expected size of the protein can be judged here. A full WB image including a control should be added here.

 Suppl table 4 what kind of genes are belonging to MSTRG numbered ones. Is the gene ID, if so use uniform annotation if not and novel transcripts then the pathway analysis is likely to be run on much fewer genes not as it was stated. See the corresponding parts in line 331.

The table does not contain EGR1 IGF and other listed genes. How were they obtained, the MGAT5 showed a -9 fold change, although it was explained as an increase. This should be clarified. 

Round 2

Reviewer 1 Report

I would like to thank the authors for addressing the concerns and suggestions. It has helped to understand the manuscript even better.

Thank you for providing the original western blot images and including discussions about Ghanim et al's work.

Author Response

We want to thank this reviewer again for all his/her constructive comments, which helped improve the quality and readability of our manuscript. We emphasized the insightfulness of his/her earlier comments on testing our RNA-based therapeutics (ASOs and CRISPR/CasRx) using SFT patient-derived primary cells, which will certainly be adopted in our future studies.

Reviewer 2 Report

Authors have responded well to all the raised comments. I would have appreciated if they were able to the PDX treatment with ASO atleast  however, I understand the limitation of carrying this and they have mentioned the same in the revised manuscript but would like to see the results upon treatment to validate the findings. Overall other changes have been incorporated as per the raised comments. 

Author Response

We thank this reviewer again for his/her insightful comments and are pleased that he/she found all our responses satisfactory. We agree with the reviewer and our future studies will test the efficacies of our NAB2-STAT6 fusion-targeting ASOs or CRISPR/CasRx using SFT PDX (patient-derived xenograft) models, which phenocopy the original SFTs better than our engineered xenograft models. As this reviewer had observed, we included relevant discussions in our revised manuscript and also want to emphasize that with our collaborators we recently established such a PDX model [1]. Encouraged by the results presented in this manuscript, future studies will involve redesigning the fusion-targeted treatments for the patient-specific fusion type and PDX studies.

References

  1. Mondaza-Hernandez, J.L.; Moura, D.S.; Lopez-Alvarez, M.; Sanchez-Bustos, P.; Blanco-Alcaina, E.; Castilla-Ramirez, C.; Collini, P.; Merino-Garcia, J.; Zamora, J.; Carrillo-Garcia, J.; et al. ISG15 as a Prognostic Biomarker in Solitary Fibrous Tumour. Cell. Mol. Life Sci. 2022, 79, 434, doi:10.1007/s00018-022-04454-4.